# Ongoing shuffling of protein fragments diversifies core viral functions linked to interactions with bacterial hosts

Bogna J. Smug [1], Krzysztof Szczepaniak[1], Eduardo P. C. Rocha[2], Stanislaw Dunin-Horkawicz[3,4] & Rafał J. Mostowy [1] ✉

Biological modularity enhances evolutionary adaptability. This principle is vividly exemplified by bacterial viruses (phages), which display extensive genomic modularity. Phage genomes are composed of independent functional modules that evolve separately and recombine in various configurations. While genomic modularity in phages has been extensively studied, less attention has been paid to protein modularity—proteins consisting of distinct building blocks that can evolve and recombine, enhancing functional and genetic diversity. Here, we use a set of 133,574 representative phage proteins and highly sensitive homology detection to capture instances of domain mosaicism, defined as fragment sharing between two otherwise unrelated proteins, and to understand its relationship with functional diversity in phage genomes. We discover that unrelated proteins from diverse functional classes frequently share homologous domains. This phenomenon is particularly pronounced within receptor-binding proteins, endolysins, and DNA polymerases. We also identify multiple instances of recent diversification via domain shuffling in receptor-binding proteins, neck passage structures, endolysins and some members of the core replication machinery, often transcending distant taxonomic and ecological boundaries. Our findings suggest that ongoing diversification via domain shuffling is reflective of a co-evolutionary arms race, driven by the need to overcome various bacterial resistance mechanisms against phages.

Modularity is a fundamental concept that pervades biological systems, shaping everything from the architecture of cells to the complexity of ecosystems[1-4]. This principle posits that biological entities are often composed of interchangeable, semi-independent units or 'modules' that serve specific functions. In evolutionary terms, modularity enhances adaptability, allowing organisms to reconfigure or expand their functional repertoires in response to changing environmental pressures[5-8].

Among the most compelling systems for studying biological modularity are (bacterio)phages—viruses that infect bacteria. David Botstein proposed as early as in 1980 that phages evolve by shuffling interchangeable functional modules within their genomes, a concept known as genomic modularity. This modular organisation facilitates the emergence of new, mosaic genotypes that are advantageous in specific niches[9]. The outcome of this genomic modularity is what is termed genetic mosaicism. Multiple studies have since demonstrated

[1]Malopolska Centre of Biotechnology, Jagiellonian University, Krakow, Poland. [2]Institut Pasteur, Université Paris Cité, CNRS UMR3525, Microbial Evolutionary Genomics, Paris, France. [3]Institute of Evolutionary Biology, Faculty of Biology & Biological and Chemical Research Centre, University of Warsaw, Żwirki i Wigury 101, 02-089 Warsaw, Poland. [4]Department of Protein Evolution, Max Planck Institute for Developmental Biology, Max-Planck-Ring 5, 72076 Tübingen, Germany. ✉e-mail: rafal.mostowy@uj.edu.pl

that genetic mosaicism is ubiquitous in phages, resulting from frequent homologous and non-homologous recombination events between different viruses[10,11]. Recombination can also occur relatively frequently between genetically distant phages[12]. As a result, bacteriophage population structure is better represented as a network rather than a phylogenetic tree[13,14], where modules of functionally related groups of genes have a coherent evolutionary history[15,16]. Following billions of years of co-evolution with bacteria, the resulting diversity of phages is astounding, a phenomenon that has only recently become fully appreciated due to advances in genomics and metagenomics[17]. Such diversity is evident not only at the genomic level—DNA[18] and RNA[19]—but also extends to the variety of phage morphologies, structures[17], bacterial resistance mechanisms[20–22] and viral counterstrategies[23,24].

While the concept of genomic modularity has been extensively studied in phages, less attention has been given to modularity within proteins, a phenomenon we refer to as protein modularity. In this context, protein modularity describes the organisation of a three-dimensional structure of a protein into distinct functional domains or units, each serving a specific role. These domains can be thought of as the building blocks of a protein, capable of being shuffled, duplicated, or modified, much like genomic modules. This can lead to domain mosaicism, which refers to the occurrence of proteins that are composed of domains or units derived from different origins[25,26]. Essentially, it serves as the manifestation of protein modularity at the level of individual proteins, akin to how genetic mosaicism serves as a manifestation of genomic modularity[27]. Hence, the occurrence of domain mosaicism can be thought of as a predictor of protein modularity.

There are compelling reasons to suspect extensive protein modularity in phages. First, certain functional classes of phage proteins, such as receptor-binding proteins (including tail fibres, tail spikes) and endolysins, exhibit remarkable modularity at both genetic and structural levels[28–31]. These modules can even be experimentally shuffled to produce viable phage virions with modified host ranges[32–35]. Second, previous studies have suggested that structural phage proteins of different functions have evolved to reuse the same folds for various purposes, with recombination being a key genetic mechanism driving this evolution[36,37]. Finally, studies that looked for the presence of composite genes (fusions of different gene families) in viral genomes found this phenomenon to be extensive[16,38]. However, the extent to which domain mosaicism occurs in phages and its relationship to genetic and functional diversity in phages has never been quantified on a large scale.

In this study, we aim to better understand the extent of protein modularity in phages and its role in viral evolution. Specifically, we analysed over 460,000 phage proteins to detect instances of domain mosaicism, defined here as two non-homologous protein sequences sharing a fragment (domain or unit). We remain agnostic as to the exact nature of the genetic process leading to this observation (e.g., genetic recombination, deletion of intergenic regions between consecutive genes, rearrangement, integration). Using a highly sensitive approach based on comparing Hidden Markov Models (HMMs) of proteins[39], we found that while domain mosaicism is widespread, some functional groups, including tail fibres, tail spikes, endolysins, and DNA polymerases, are particularly enriched in mosaic compositions.

## Results

### Functional annotation using protein fragments is often ambiguous

To investigate the relationship between protein diversity, function and modularity in bacteriophages, we carried out a comprehensive analysis of Hidden Markov Model (HMM) profiles of representative phage proteins by comparing their predicted functional annotations, genetic similarity and domain architectures (see "Methods" and Supplementary Fig. S1). Briefly, we used `mmseqs2` to cluster 462,721 predicted

protein sequences in all bacteriophage genomes downloaded from NCBI RefSeq. The clustering was carried out at 95% coverage threshold to ensure that all proteins grouped within a single cluster have an identical or near-identical domain architecture. We took 133,574 representative protein sequences from the resulting clusters and used them to query the `UniClust30` database. The alignments obtained were then converted into HMM profiles, which we will henceforth refer to as representative HMM profiles or rHMMs. To assign functions to rHMMs, we used `hhblits`[40] to search each rHMM against the PHROGs database[41] (to our knowledge the most accurate mapping to date between diverse phage proteins and manually curated functional annotations) complemented with a database of antidefence phage proteins[42]. Throughout this study, we employed HMM-HMM comparisons to align our rHMMs with themselves or other HMM databases. In HMM-HMM comparisons, two profile HMMs are aligned with each other, capturing the statistical properties of the sequence diversity in each protein family. This involves scoring the likelihood that the amino-acid patterns in one HMM profile match those in another, thereby providing a robust and highly sensitive measure of similarity between representative proteins[39].

We investigated the robustness of the PHROG functional annotations (which were additionally simplified to combine closely related biological functions; see "Methods" and Supplementary Data S1) by assessing how the HMM-HMM comparison parameters affected both the functional coverage of the data (i.e., proportion of representative proteins with any functional hit) and functional uniqueness (i.e., proportion of annotated representative proteins with unique functional hits). We found that pairwise coverage (both query and subject) had a much stronger effect on functional assignment than hit probability (see Supplementary Fig. S2). Specifically, while changing the coverage threshold from 80% to 10% (while maintaining a high probability threshold of 95%) increased the functional coverage from 19% to 34%, it also decreased functional uniqueness from 93% to 52%—meaning that at the lowest coverage threshold every second, annotated rHMM had multiple, different functional assignments. We also found that at high pairwise coverage threshold ambiguous functional assignment often reflected biological similarity (e.g., ribonucleoside reductase vs. ribonucleotide reductase, or transcriptional regulator vs. transcriptional activator; see Supplementary Fig. S3). By contrast, at lower sequence coverage thresholds co-occurrences between clearly different functions became more and more common and affected the majority of functions (see Supplementary Fig. S3), meaning it was often impossible to confidently assign a function based on a fragment of a protein (i.e., partial match to a reference database; here PHROGs). Considering this, for further analyses we set the probability and pairwise coverage cut-offs for the PHROG annotations to 95% and 80%, respectively, while conservatively excluding all rHMMs with hits to more than a single functional class (see "Methods").

### Proteins assigned to different functional classes share homologous domains

Given that for a low pairwise coverage threshold we often found rHMMs to be co-annotated by apparently distinct functional classes, we hypothesised that these functions contained rHMMs that shared homologous domains (i.e., protein structural and functional units that have been shown to have emerged from a common ancestor). To address this hypothesis, we used the Evolutionary Classification of Protein Domains (ECOD) database as it provides a comprehensive catalogue of known protein domains and their evolutionary relationships[43]. We then used HMM-HMM comparisons to detect the presence of these domains in rHMMs (see "Methods" and Supplementary Fig. S1). The ECOD database categorises protein domains based on their evolutionary relationships and structural similarities, organising them into a hierarchical system which includes possibly homologous groups (X), homologous groups (H) and topologies (T).

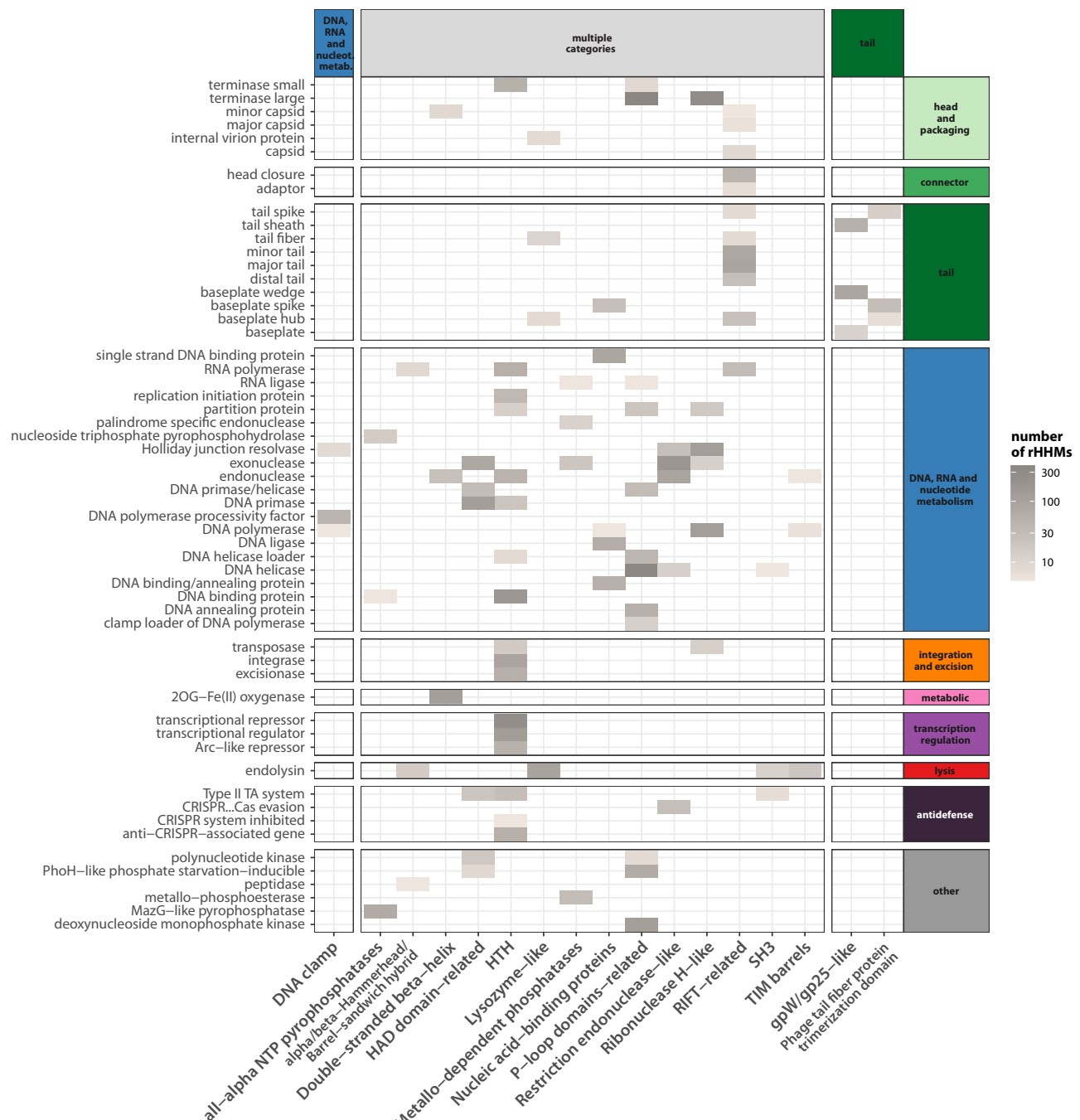

**Fig. 1 | Diverse protein functions often share homologous domains.** Heatmap showing groups of homologous ECOD domains (H-group names; X-axis) found in proteins assigned to different functional classes (Y-axis). A domain was considered present in a functional class when it was present (i.e., found with a minimum 95% probability and 70% subject coverage) in at least 5 rHMMs assigned to a given functional class. The colour scale indicates number of rHMMs in which the domain was found. Only domains found in multiple functional classes (at least 3) are shown. Generic functional classes (tail and structural protein) were excluded from this visualisation. Functional classes are grouped and coloured according to their categories. Source data are provided as a Source Data file. rHMM: representative HMM profile.

Domains with a similar spatial arrangement of their secondary structures are assumed to share the same topology and are assigned to the same T-group. Domains from different T-groups are grouped into homologous groups (H-groups) when there is evidence of their homology via sequence similarity, functional similarity or other features. Domains that lack sequence similarity but share structural features can still be classified as possibly homologous and be assigned to the same X-group. We thus assumed that domains in different X-groups are unrelated.

Figure 1 shows the distribution of ECOD domains (H-groups) across different functional classes. (The distribution of T-groups across functional classes is also shown in Supplementary Fig. S4.) In line with previous literature, we found examples of phage proteins with different functions sharing homologous domains. These include well-known examples of helix-turn-helix domains found in transcriptional regulator/repressor proteins, integrases, transposases or DNA-binding proteins[44]; the RIFT-related domains found in many structural proteins like tail and neck proteins[45] but also for example in RNA

polymerases in the form of double-psi barrels[46]; P-loop domain-related family found in ligases, kinases or helicases[47–49]; or the ribonuclease H-like domain family found in many DNA processing enzymes like Holliday junction resolvases, exonucleases, DNA polymerases or transposases[50].

Cases of homologous domain sharing (i.e., belonging to a single ECOD H-group) between proteins assigned to different functional classes can be explained in several ways. One explanation is that such proteins may actually have the same function (e.g., baseplate and baseplate wedge). Alternatively, ancient and large domain classes that play an important, biological role (e.g., DNA binding or NTP hydrolysis) may have diverged into subfamilies specific for different functions and thus are shared by a wide range of PHROG classes. Indeed, we found a strongly significant, positive correlation between domain frequency (number of rHMMs containing each H-group) and diversity (number of predicted ECOD families within each H-group; see Supplementary Fig. S5), suggesting that domains that are common in nature tend to be more diverse.

Finally, domain sharing between proteins assigned to different functional classes may be the result of mosaicism, i.e., the acquisition of specialised domains for different functions. This scenario is additionally supported by the observation that distinct H-groups were detected in proteins assigned to the same functional class. For example, within proteins assigned as exonucleases, endonucleases, DNA polymerases or endolysins we found as many as 4 distinct H-groups, each found in at least 5 rHMMs (although not necessarily all together in the same rHMM; see Fig. 1). We thus hypothesised that this distribution is indicative of modularity and ongoing domain shuffling in functionally diverse proteins.

## Protein modularity is most often linked to replication, lysis and structural proteins

To detect and quantify domain mosaicism and better understand its relation to protein function, we studied the presence or absence of ECOD domains within pairs of proteins. Specifically, a pair of proteins (represented by rHMMs) was considered to exhibit domain mosaicism if it met the following criteria (see also Fig. 2A): (1) each protein in the pair has at least two domains belonging to different X-groups, (2) the pair shares at least one domain with the same topological features (i.e., belonging to the same T-group), and (3) each protein in the pair also contains at least one domain with unique structural architecture (i.e., belongs to an X-group absent in the other protein). We refer to this as ECOD-based mosaicism. We found evidence of such mosaicism in 45 out of 101 functional classes (assuming at least three mosaic rHMMs per functional class). Figure 2B shows a map of ECOD-based mosaicism visualised as a network with nodes representing functional classes and edges linking those classes that contained rHMMs with evidence of ECOD-based mosaicism. We found that functional categories where domain-level mosaicism was common were DNA/RNA metabolism (e.g., RNA and DNA polymerases, DNA ligases, helicases, exo- and endonucleases, DNA-binding proteins), transcription regulation, structural tail proteins (tail fibre, tail spike and baseplate proteins) and endolysins. Three functional classes with the most examples of within-class ECOD-based mosaicism were DNA polymerases, endolysins and tail spikes.

To examine the relationship between protein function and ECOD-based mosaicism independently of assignment to functional classes, we next investigated which domain architectures are statistically associated with such mosaicism. To this end, for each domain (ECOD T-group), we calculated the odds ratio of being over-represented in proteins with evidence of ECOD-based mosaicism vs. those without any evidence of ECOD-based mosaicism (see "Methods" section). Specifically, we first considered only rHMMs with significant hits to at least a single domain. Then, for a given domain, we calculated the number of all domain architectures (i.e., unique combinations of ECOD

T-groups) with and without that domain and the number of all domain architectures with and without evidence of ECOD-based mosaicism. Finally, we calculated the odds ratio that this domain is found more frequently in mosaic domain architectures than non-mosaic domain architectures (see "Methods"). Results, shown in Fig. 2C, are consistent with the network in panel B. Domains with the greatest odds ratio of being over-represented in mosaic proteins typically fall into three categories: (1) domains occurring in proteins associated with DNA/RNA metabolism, particularly in DNA polymerases, DNA primases, DNA helicases, exonucleases, ribonucleotide reductases and Holliday junction resolvases, e.g., P-loop containing nucleoside triphosphate hydrolase, Ribonuclease H-like, adenylyl and guanylyl cyclase catalytic, toprim or SAM-like domains; (2) domains occurring in endolysins, e.g., lysozyme-like, SAM-like, cysteine proteinases or SH3; and (3) domains occurring in receptor-binding proteins, e.g., pectin lysase-like or tail fibre trimerization domain.

The existence of domain mosaicism in phages is not a new phenomenon as some functions analysed here have been previously linked with mosaic domain architectures[51,52]. We thus next enquired which cases of ECOD-based mosaicism are ancient (i.e., represent ancestral domain shuffling underlying functional diversification) and which cases of ECOD-based mosaicism are contemporary (i.e., are the result of an evolutionarily recent reshuffling of domains). This issue was partially addressed using ECOD T-groups instead of H-groups to assign shared domains in protein pairs. However, to investigate this problem further, we first looked into the sequence similarity distribution of all mosaic pairs of rHMMs and found that only 9% of them shared fragments with a percentage identity of 10% or greater. Then, we reanalysed the data using a definition of contemporary mosaicism by requiring that the shared protein fragments have an amino-acid percentage identity level of 50% or greater. We found that four of the functional classes fulfil that criterion (Fig. 2B, brown edges): DNA polymerase, tail spikes, endolysins and tail fibres. Finally, using the domain-based approach (Fig. 2D), we found that domains (ECOD T-groups) significantly over-represented in proteins showing evidence of contemporary mosaicism are most often linked to receptor-binding proteins and baseplate proteins (e.g., putative tailspike protein N-terminal domain, pectin lyase-like, N-terminal Ig-like domain) and to endolysins (e.g., LysM domain, SH3, amidase-like, lysozyme-like etc.); we also found a signal to domains that are typically associated with replication (HAD domain-related and P-loop domains-related). (The full list of odds ratios and p-values from Fig. 2C and D can be found in Supplementary Data S2.)

Overall, these results show that (1) domain mosaicism, measured with ECOD-based mosaicism, is common in phage proteins and associated with DNA/RNA replication, lysis and structural proteins, (2) while most of that mosaicism appears to be due to ancient domain shuffling or specialisation, we see clear examples of contemporary mosaicism particularly in receptor-binding proteins and endolysins, and (3) there are also rare and intriguing cases of recently emerged mosaicism associated with other functions.

## Domain mosaicism hotspots

To better understand the nature of protein modularity, we next looked into the specific domain architectures of the four mosaicism-outliers: DNA polymerase, tail fibre, tail spike and endolysin. We also developed a Shiny webserver that allows users to interactively look up and visualise domain architectures in all functional classes used in this analysis as well as to connect specific domains shared with proteins of other functions: bognasmug.shinyapps.io/PhageDomainArchitectureLookup.

**DNA polymerase and other replication proteins.** As far as individual functional classes are concerned, DNA polymerases are the clear mosaicism outlier in the 'DNA, RNA and nucleotide metabolism' category. The representative domain architectures of all of those found in

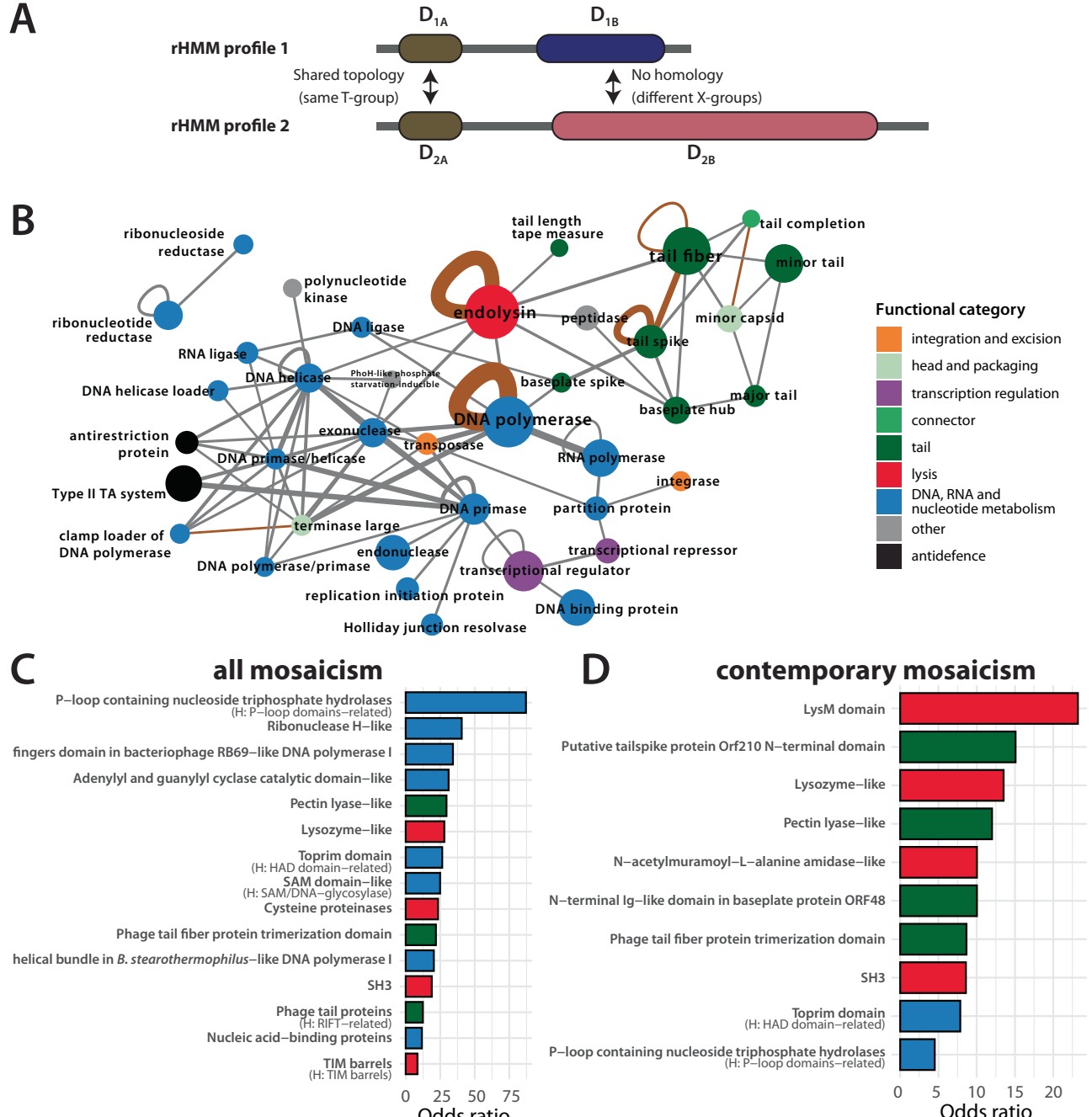

**Fig. 2 | Map of ECOD-based mosaicism in phages. A** ECOD-based mosaicism for any rHMM pair was defined when (1) both proteins had at least two distinct ECOD domains detected, (2) both shared a domain assigned to the same ECOD T-group and (3) both additionally contained non-homologous domains (i.e., belonging to different X-groups). **B** Mosaic network of protein functions. Each node represents a functional class and edges link functional classes where evidence of domain mosaicism was found between at least four pairs of domain architectures (i.e., unique combinations of ECOD T-groups which can be thought of as structurally equivalent proteins). Brown edges connect functions where at least one case of contemporary mosaicism was found (i.e., a pair of rHMMs with the percentage identity of a shared fragment 50% or greater). Node size corresponds to the number of domain architectures in a given functional class. Edge thickness corresponds to the number of domain architecture pairs with evidence of domain mosaicism.

Generic functional classes (tail and structural protein) were excluded from this visualisation. **C** Bar plot shows the odds ratio that a given domain (ECOD T-group) is found more frequently in mosaic domain architectures than non-mosaic domain architectures. Only domains for which the results were statistically significant are shown (Fisher's one-tailed exact test; significance level was set at $p < 0.05$, with $p$-values adjusted using Bonferroni correction for multiple testing, $n = 1011$ domain architectures). Colours denote the most frequent functional category in rHMMs with the given domain. For each domain, corresponding H-group names are provided if different from the T-group name. **D** Same as panel (**C**) but here mosaicism is defined as contemporary as for brown edges in panel (**B**). The full list of odds ratios and exact $p$-values from panels (**C** and **D**) can be found in Supplementary Data S2 ($n = 1011$ domain architectures). Source data are provided as a Source Data file. rHMM: representative HMM profile.

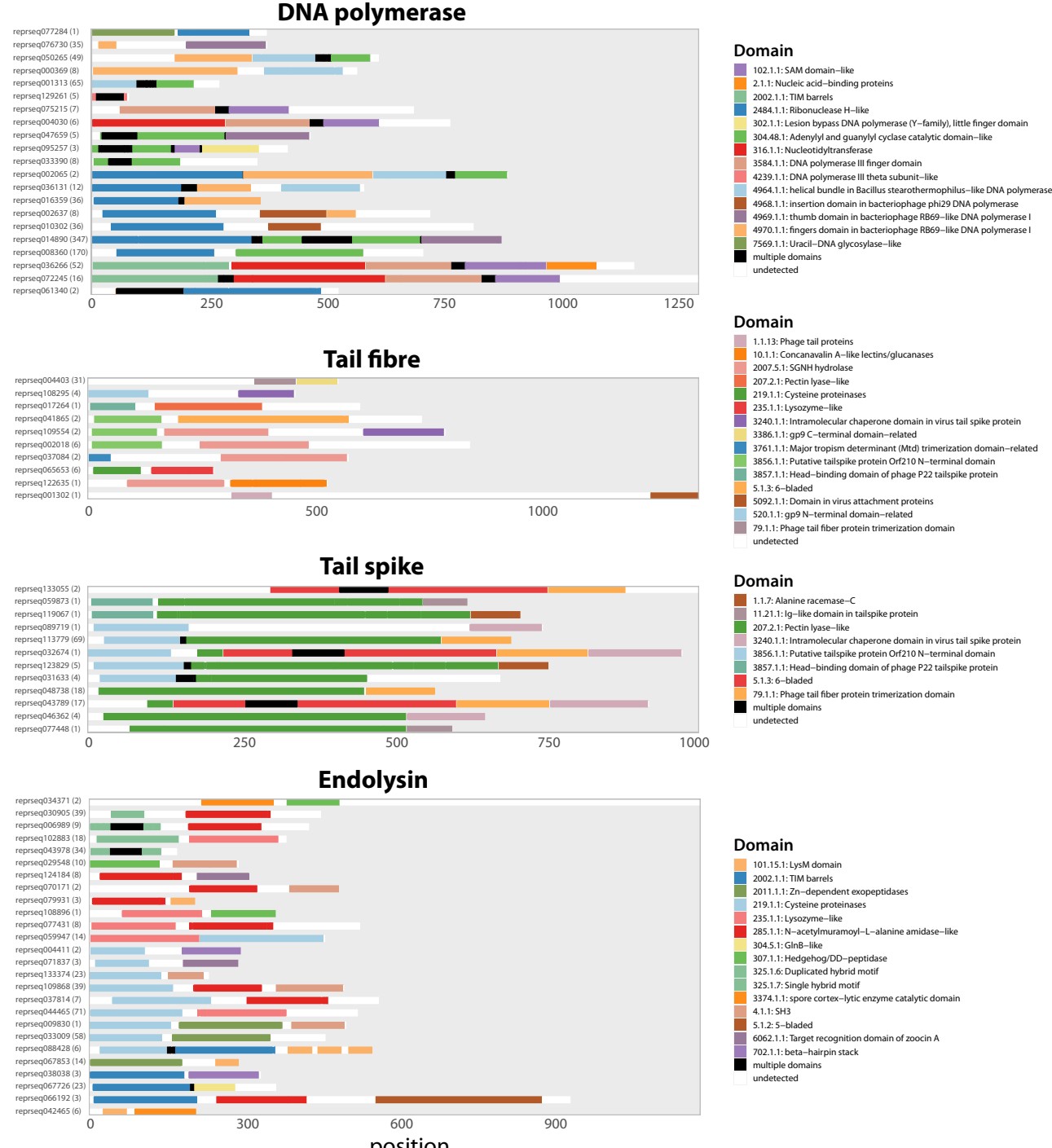

**Fig. 3 | Visualisation of domain architectures for functional classes exhibiting the highest levels of ECOD-based mosaicism.** Each line shows one chosen rHMM (abbreviated `reprseqXXXXXX`) per each domain architecture, with the number of protein sequences having this domain architecture displayed in bracket, and the ECOD domains (T-groups) found within that rHMM. Colours denote the ECOD T-groups, with black denoting multiple domains found in this region and white denoting the absence of ECOD hits. ECOD T-groups are in the following format: $X_{id}. H_{id}. T_{id}$. Only domain architectures with at least two different T-groups are shown. Source data are provided as a Source Data file. rHMM: representative HMM profile.

DNA polymerases are shown in Fig. 3. For comparison, in Supplementary Fig. S6 we show an overview of the domain architectures for representative members of all families of DNA polymerases known to occur in bacterial or viral genomes (A, B, C, X and Y)[53,54] detected using `HHpred`[55] with ECOD as the database.

Our results point to a few notable observations. First, we have recovered domain architectures of not only families A and B, which are well known to occur in phages such as T4 and T7 but also of families C and Y (c.f., Supplementary Fig. S6) which are characteristic of bacteria. Second, we have identified other domain architectures that are variants of the above. For example, instead of the four domains typical of family A (ECOD X-groups 2484, 4970, 4964 and 304), we found rHMMs that contained only the first three (2484, 4970, 4964) and two (2484, 4970). Such rHMMs with unusual domain architectures represented

clusters with multiple protein sequences, suggesting multiple occurrences of such architectures in the analysed genomes. Finally, the comparison of these domain architectures points to clear cases of mosaicism, such as the insertion domain of bacteriophage $\phi$29 found alongside the exonuclease (ECOD X-group 2484) and/or the finger domain (ECOD X-group 4970). It also highlights that conserved folds found in DNA polymerases are reused in various combinations, but also in combination with domains present in other proteins belonging to the 'DNA, RNA and nucleotide metabolism' category (see Supplementary Fig. S7).

**Receptor-binding proteins and other tail proteins.** Receptor-binding proteins, like tail fibres and tail spikes, are often described in the literature as consisting of three domains: a conserved N-terminal that binds to the tail structure (e.g., to the baseplate), a variable and host-dependent C-terminal that binds to the receptor at the bacterial surface, and the central domain which contains enzymes (hydrolases) that help the phage penetrate layers of surface sugars like the capsular polysaccharide[56]. Our results show clear evidence for the emergence of domain mosaicism via shuffling of all of these domains (see Fig. 3). First, we find N- and C-terminal domains in multiple arrangements. For example, the C-terminal 'Alanine racemase-C' domain (ECOD T-group 1.1.7) is found in tail spikes in combination with either the 'Head-binding domain of phage P22 tailspike protein domain' (3857.1.1) or with the 'Putative tailspike protein Orf210 N-terminal domain' (3856.1.1), providing an excellent example of mosaicism. Second, we found the co-existence of various enzymatic domains within the same protein in different combinations. For example, the 'Intramolecular chaperone domain in virus tail spike protein' domain (3240.1.1) was found to co-occur with the SGNH hydrolase domain (2007.5.1) in a tail fibre protein as well as with the Pectin lyase-like domain (207.2.1) in a tail spike. Finally, some domains present in receptor-binding proteins were also found to occur in other functional classes. A good example here is the tail fibre trimerization domain (79.1.1) which is also found in baseplate spikes in combination with other domains like lysozyme (see also Supplementary Fig. S8). Overall, these results suggest that domains found in receptor-binding proteins can not only be shuffled in different combinations, but that multiple enzymatic domains can co-occur in the same protein.

**Endolysins.** Endolysins are classically described as having catalytic domains (lysozymes, muramidases, amidases, endopeptidases, etc.) and/or cell wall-binding domain; and they may be observed in multiple combinations[29,30,57]. Here, we find both types of domains co-occurring in various combinations (see Fig. 3). For example, the catalytic domain Cysteine proteinases (ECOD T-group 219.1.1) is found in combination with either the SH3 domain (4.1.1) or target recognition domain of zoocin A domain (6062.1.1). We also found the presence of multiple catalytic domains within the same proteins. This includes the co-occurrence of exopeptidases (2011.1.1) and Cysteine proteinases (219.1.1), or co-occurrence of the spore cortex-lytic enzyme (3374.1.1) and Hedgehog/DD-peptidase (307.1.1). These domain architectures are in line with those previously described for endolysins of mycobacteriophages, where apart from multiple instances of co-occurrence between the peptidase-like N-terminal and a cell wall-binding C-terminal there were also central domains with amidases, glycoside hydrolases and lytic transglycosylases[57]. Domain architectures composed of multiple catalytic domains, and sometimes lacking any cell wall-binding domains have also been reported by Criel and colleagues[29].

Interestingly, a domain-based network of diversity in all lysis genes exhibited a higher level of interconnectedness than for replication and tail protein networks (see Supplementary Fig. S9). This phenomenon suggests that endolysin domains likely co-occur in many out of all theoretically possible combinations, which is consistent with previous analyses of domain diversity in endolysins[29,31,57].

## Sequence-based insight points to extensive mosaicism beyond domain analysis

ECOD-based mosaicism—our measure of domain mosaicism—serves as a robust approach for detecting protein modularity. It uses the evolutionary information from the ECOD database to discern whether a lack of local sequence similarity is due to evolutionary divergence (same T-groups) or due to a lack of common ancestry (different X-groups). The approach also ensures that the units of mosaicism (i.e., domains) are evolutionarily meaningful as they can fold independently and hence can be horizontally shuffled. However, this approach has two important limitations.

The first limitation is that the approach to detect domain mosaicism relies on the assumption that all functional classes have a comparable coverage in the ECOD domain database, which may not be true. Indeed, our analysis of such coverage (see Supplementary Fig. S10) shows that while some functional classes—for example, those belonging to the 'DNA/RNA nucleotide metabolism' category—are relatively well annotated by ECOD, other functional classes (e.g., tail completion, head scaffolding, spanin, holin/anti-holin, nucleotide kinase or tail length tape measure) seem to be strongly underrepresented in domain databases like ECOD. An interesting example is tail fibres, which rarely exhibit hits to more than a single ECOD domain in spite of being known as long and multi-domain proteins. Furthermore, while we saw a strong and significant correlation between structural diversity (number of unique domain architectures detected by ECOD at the T-group level) and genetic diversity (measured by the number of protein families, where protein family was defined as a cluster of similar rHMMs; see "Methods") in different functional classes, some classes—including tail length tape measure protein, membrane proteins, head-tail joining proteins or ssDNA binding proteins—had a disproportionately low structural diversity compared to genetic diversity (see Supplementary Fig. S11).

The second limitation of the ECOD-based approach to detect domain mosaicism is that it relies on a highly restrictive definition of mosaicism—it requires that each protein in a mosaic pair has two structurally unrelated domains detected (different X-groups). This requirement might miss many cases of mosaicism where a domain is undetected or when mosaicism occurs at the sub-domain level[58]. To gauge the potential extent of such bias, we carried out the all-against-all comparison of 134k rHMMs using `hhblits`[40] and investigated the relationship between their sequence similarity and coverage of all pairs (see Supplementary Fig. S12A). The results show that, while most rHMM pairs with any detected similarity align at high coverage, reflecting their likely homology over the majority of the sequence length, there is a substantial fraction of pairs that overlap by a fragment that constitutes a short proportion of their length, indicating possible mosaicism at the domain or sub-domain level. The number of pairs that overlap on a short proportion of length was nearly the same when we subtracted all rHMM pairs where we detected ECOD-based mosaicism (Supplementary Fig. S12B), suggesting that this measure of domain mosaicism is highly restrictive and hence imperfect.

Given those limitations, we introduced another measure of domain mosaicism, hereafter called sequence-based mosaicism, aimed at finding instances of similar sequences shared by otherwise unrelated proteins (see Fig. 4A). To detect sequence-based mosaicism, we queried all rHMMs against each other using HMM-HMM comparisons. Then, for every rHMM pair for which similarity was detected, we calculated query and subject coverage as the total number of residues in the aligned sequence regions divided by the length of the query and subject sequence, respectively. Finally, for each pair, we calculated the pairwise coverage as the maximum of subject and query coverage, and pairwise probability as the average of HHsuite probabilities assigned to each residue in the homologous region. A pair of rHMMs was designated as mosaic according to the sequence-based definition if they exhibited detectable similarity across less than half of their respective

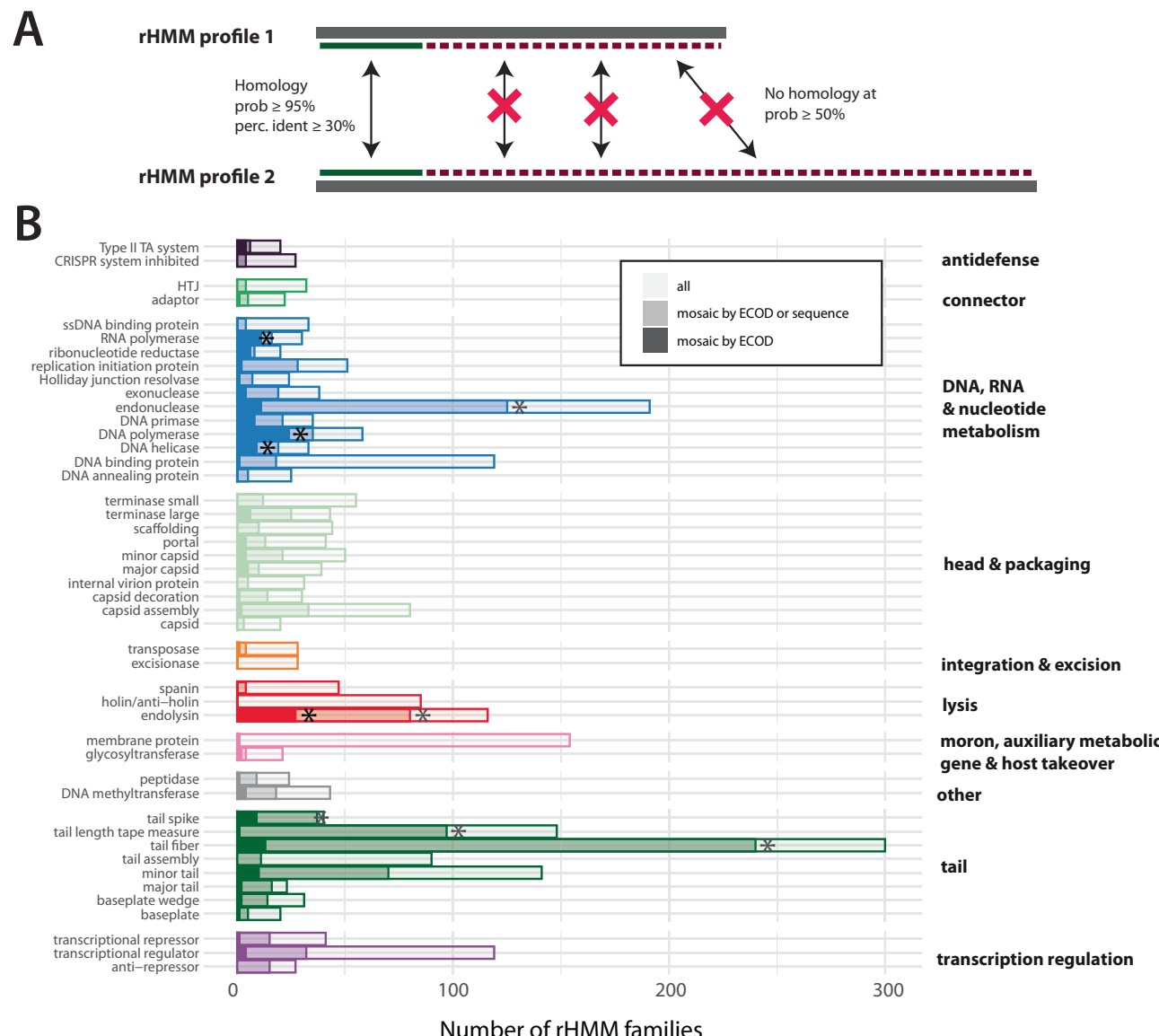

**Fig. 4 | Comparing measures of domain mosaicism. A** Sequence-based mosaicism for any rHMM pair was defined, using comparison of their HMM profiles, as presence of sequence-based similarity in the background of absence of homology. Pairwise coverage was calculated as the total aligned length using a permissive probability hit threshold of 50%. Sequence mosaicism was claimed when the aligned fragment (1) constituted a short proportion of the length of both proteins (both query and sequence coverage ≤50%), (2) was detected with high probability ($p ≥ 95\%$), (3) had percentage identity of at least 30% and (4) was of length ≥50aa. **B** Number of rHMM families with signal of mosaicism detected by ECOD-based definition alone (opaque) vs. by ECOD-based or sequence-based definition (moderate) vs. number of all rHMM families (transparent) in different functional classes. Colours are equivalent to those in Fig. 2B. Only functional classes with genetic diversity of at least 20 families are shown. Functional classes that were significantly over-represented in mosaic proteins (using ECOD-based definition or either definition) are marked with an asterisk. Over-representation in each functional class was tested using Fisher's one-tailed exact test, and $p$-values were adjusted for multiple testing using Bonferroni correction ($n = 3078$ rHMM families). The full list of odds ratios and exact $p$-values is available in Supplementary Data S3. Source data are provided as a Source Data file. rHMM: representative HMM profile.

lengths (with a pairwise coverage of 50% or less) but with a minimum length of 50 amino-acids, minimum of 30% identity in the aligned protein region and a hit probability of at least 95%. Furthermore, any rHMM involved in such a mosaic pair was termed an rHMM with a signal of sequence-based mosaicism.

Figure 4B shows the number of rHMM families with a signal of mosaicism detected by ECOD (i.e., ECOD-based mosaicism) vs. by ECOD or sequence (i.e., ECOD-based mosaicism or sequence-based mosaicism) in different functional classes. We saw that, on average, the proportion of rHMM families with the signal of sequence-based mosaicism in a given functional class was greater than the proportion of rHMM families with a signal of ECOD-based mosaicism. This result

was in line with our expectations since, as explained above, the sequence-based mosaicism approach is the less conservative one. Notably, however, in some functional classes, the proportion of families with a signal of sequence-based mosaicism alone was disproportionately high. Examples include functional classes such as replication initiation protein, endonuclease, DNA-binding protein, capsid assembly, endolysin, tail spike, tail length tape measure, tail fibre, minor tail or transcriptional repression/regulation. In particular, some of the functional classes that were not over-represented in rHMMs mosaic according to ECOD-based definition turned out significantly over-represented when we used the sequence-based definition. These include tail fibres, tail spikes, tail length tape measures and

endonucleases. (The full list of odds ratios and *p*-values is available in Supplementary Data S3.) These observations suggest that these—and potentially some other—functional classes may harbour an under-explored reservoir of domain mosaicism.

## Protein diversification via domain shuffling is an ongoing evolutionary process

Our results so far indicate that domain mosaicism frequently underlies functional diversity in phages. In some functional classes, such as receptor-binding proteins (RBPs) and endolysins, the emergence of domain mosaicism can be considered 'contemporary,' based on our own definition that restricts the term to cases where the percentage identity of the shared fragment is at least 50%. Furthermore, using a sequence-based definition of mosaicism—with a baseline of percentage identity threshold of 30%—we found that proteins assigned to multiple functional classes exhibit mosaicism that is not detected by the ECOD-based analysis. However, these thresholds may still permit cases of domain mosaicism that are relatively old—particularly if the shared fragments are evolutionarily conserved. Therefore we were unable to reveal which functions, if any, may be subject to ongoing, rapid diversification. To provide a more nuanced view, we introduce a category 'recently emerged mosaicism' to classify cases of recently emerged sequence mosaicism using two levels: high-confidence (percentage identity ≥ 70%) and very high-confidence (percentage identity ≥ 90%). We then created a network of functional classes with rHMMs that exhibit a signal of a recently emerged mosaicism at both confidence levels (see Supplementary Fig. S13). While, as expected, multiple links were found between proteins classified as tail fibres, tail spikes and endolysins, connections between proteins of other functions were also identified, including replication proteins, neck proteins and anti-defence proteins. Additionally, almost all of the functional classes linked with rHMMs with an unknown status, implying that they had a mosaic signal with a protein the function of which was uncertain.

To investigate whether instances of recently emerged mosaicism are genuine indicators of ongoing protein diversification rather than false positives, we employed a multi-faceted approach. This involved pairwise comparisons, `HHpred` domain detection and genomic context analysis to examine dozens of these pairs in detail. Consequently, we present illustrative examples from six different functional classes: neck passage protein, tail fibre, endolysin, ribonucleotide reductase, replication initiation protein, and DNA polymerase (see Fig. 5). These examples serve as representative cases for their respective functional classes, showcasing the diverse mechanisms and biological contexts in which domain shuffling facilitates ongoing protein evolution.

As shown in Fig. 5A, one mechanism that mediates protein diversification is the exchange of domains. This can be best seen using the example of neck passage structure proteins. These proteins have been previously identified as a diversity hotspot in *Lactococcus* phages[59] and some are known to carry carbohydrate-binding domains[60]. The provided example shows an exchange of a non-homologous C-terminal receptor-binding domain and pectin lyase-like domain while preserving the near-identical N-terminal in two closely related phages. An analogous example are two tail fibre proteins (Fig. 5B), found in closely related *Klebsiella* phages, with a very similar N-terminal and two, non-homologous receptor-binding C-terminal domains—a phenomenon very well known to occur in phages infecting bacteria with extensive surface polysaccharide diversity[28]. A similar but converse example are two fragment-sharing endolysins found in two otherwise unrelated genomes of *Anoxybacillus* and *Aeribacillus* phages (Fig. 5C). The said endolysins contain a highly similar C-terminal (lysozyme) and two unrelated N-terminal domains (exopeptidase and amidase).

On the other hand, we observed multiple different mechanisms driving protein diversification in core replication proteins. One was domain exchange between ribonucleotide reductases in *Lactococcus*

phages (Fig. 5D). The two closely related genomes both carry a ribonucleotide reductase protein with an identical N-terminal domain and unrelated C-terminals: ten stranded beta/alpha barrel domain (ECOD 2500.1.1) and FAD/NAD(P)-binding domain (2003.1.2). Interestingly, one of the genomes has the other C-terminal domain in another protein that is located downstream from a genetic island that contains other ribonucleotide reductases and endonuclease domains. This suggests that diversification of the discussed protein was linked to the insertion/deletion of a new domain, possibly together with the mentioned genetic island.

Another example of a protein diversification mechanism was found in two replication initiation proteins present in two closely related genomes of *Gordonia* phages (Fig. 5E). The two proteins share near-identical C-terminal regions but with no detectable ECOD domain hits; they also both have hits to the winged helix-turn-helix domain (ECOD 101.1.2) but with no detectable similarity at the sequence level. While the homologous N-terminal could potentially be explained by strong diversifying selection, the high similarity between the two phage genomes (ANI = 97%, coverage = 89%) suggests that the most likely explanation is a domain exchange via recombination into its distantly related variant.

Last but not least, we investigated the underlying mechanism of diversification of DNA polymerases. Interestingly, this mechanism is quite different from the ones above and involves shuffling (i.e., gain or loss) of domains, as shown in Fig. 5F. Two proteins, found in related genomes of *Bacillus* phages, share an identical sequence that we identified as a helical bundle in DNA polymerase I. Investigation of other proteins in the neighbouring genetic region revealed that the two genomes contain the same set of DNA polymerase domains at a high percentage identity but split into different open reading frames due to the presence and absence of several endonucleases between those domains. This suggests that the diversification of replication regions, including DNA polymerases, in phages may often occur via the gain and loss of domains.

## Domain mosaicism transcends taxonomic and ecological boundaries

We next sought to determine whether domain mosaicism is restricted to specific taxonomic and ecological groups of phages. To address this, we assigned taxonomic information (family and genus), host (at the bacterial genus level) and predicted lifestyle (temperate or virulent) to all relevant genomes in the NCBI RefSeq database (see "Methods"). For each genome, we then calculated the proportion of proteins with signal of mosaicism (ECOD-based or sequence-based). This analysis yielded several intriguing observations.

First of all, as shown in Supplementary Fig. S14, our results suggest that temperate phages possess a higher proportion of proteins with signal of mosaicism compared to lytic phages. These findings are in line with the existing literature on the greater genetic mosaicism and frequency of horizontal gene transfer in temperate phages compared to virulent ones[17]. Additionally, the observed differences in proportions of proteins with signal of mosaicism between phage families (see Supplementary Fig. S15) and host genera (see Supplementary Fig. S16) were statistically significant, with *Autographiviridae* phages typically having shorter genomes and showing a higher proportion of proteins with signal of mosaicism compared to other families (see Supplementary Fig. S15). However, a comparison of the proportion of proteins with signal of mosaicism between multiple groups of phages should be interpreted with caution due to potential confounding factors, most importantly due to the limited ability to detect mosaicism in proteins from phages that are under-represented in databases.

We further investigated whether specific phage taxonomic or ecological groups predominantly contribute to the observed domain mosaicism in certain functional classes such as tail spikes or fibres, DNA polymerases or endolysins. Our analysis revealed that signals of

**Fig. 5 | Domain mosaicism caught red-handed.** Six panels (**A**–**F**) show representative examples of recently emerged domain mosaicism. (Left) Horizontal lines denote the protein sequence length (abbreviated `reprseqXXXXXX`) with dashes every 100aa. Blue shows the region with high genetic similarity and information about the percentage identity of that region (amino-acid level). Boxes show detected ECOD domains with their ECOD IDs; homologous domains in a pair have the same colour. (Right). Genomic comparison of the regions where the two proteins were found, with the corresponding names of the phage and genome coordinates. The upper genome fragment corresponds to the upper protein, and stars show the location of the two proteins. Only proteins with informative functional hits (NCBI Genbank) are labelled and marked in green; otherwise, they are grey. Links are drawn between genes with percentage identity of at least 30% across the full length, with the level of identity represented by the scale at the bottom of the figure. Genome comparison visualisations were made using `clinker`[97], and the corresponding NCBI accession numbers are displayed.

mosaicism were widespread across multiple phage families (see Supplementary Fig. S17). For instance, high levels of mosaicism in tail spikes were primarily observed in the *Kuttervirus* genus, while mosaicism in proteins related to DNA metabolism was more prevalent in phages with larger genomes, such as *Straboviridae* and *Hellerviridae*. Unsurprisingly, when categorising phages as either lytic or temperate, we found that certain functional classes, like those related to integration or transcription regulation, were almost exclusively found in temperate phages (see Supplementary Fig. S18). Moreover, endolysins tended to be more mosaic in temperate phages, whereas tail fibres and DNA polymerases exhibited greater mosaicism in lytic phages (see Supplementary Fig. S18).

Finally, we examined whether domain mosaicism transcends taxonomic or ecological boundaries. Our analysis revealed that for not-recently-emerged mosaicism (percentage identity < 70%), rHMMs in a mosaic pair often belonged to different groups (e.g., temperate and virulent) or consisted of proteins conserved between the groups (see Supplementary Fig. S19). In contrast, recently emerged mosaicism (percentage identity ≥ 70%) typically involved rHMMs from the same group, although in some cases they came from different genera or hosts, and more rarely, from different taxonomic families or lifestyles (see Supplementary Fig. S19). Notably, recently emerged mosaicism involving rHMM pairs from distinct taxonomic families or lifestyles was exclusively observed within receptor-binding proteins, specifically tail fibres and tail spikes (see Supplementary Fig. S20).

## Discussion

In this study, we have systematically analysed the relationship between genetic diversity, functional diversity and protein modularity (detected by instances of domain mosaicism) in phages using 134k Hidden Markov Model (HMM) profiles of representative phage proteins (rHMMs). We compared these profiles to each other and to the ECOD domain database using a sensitive homology search via HMM-HMM comparison. While alternative methods for comparing distant sequences of proteins exist[61,62], HMM-to-HMM alignments are known to be the most accurate, able to detect homology even when the sequence similarity falls below 10%[39,63]. Even though there have been recent emerging techniques—such as those employing natural language processing[64]—that show promise in achieving sensitivities comparable to HMM-HMM methods, they have not yet undergone the extensive validation that profile-profile alignment methods have. Therefore, we opted for the well-established and highly reliable HMM-HMM comparison approach to ensure the robustness of our findings.

Our results demonstrate that domain conservation in phage proteins is extensive, often linking proteins with different functions and that these domains often co-occur in multiple combinations. This is consistent with our knowledge of how phages evolve and their remarkable ability to not only alter their protein sequence through rapid evolution but also to recycle existing folds in novel biological contexts[36,37,65]. Indeed, our findings show that such domain shuffling, which is known to be often recombination-driven[66], not only links different functions but also underlies genetic diversity within multiple functional classes, notably related to tail proteins, lysins, and the core replication machinery. Our results also demonstrate that domain shuffling is a vital mechanism for ongoing diversification in phage populations. The recent emergence of domain mosaicism in proteins from phages of varied taxonomic and ecological origins underscores the pivotal role this mechanism plays in facilitating viral adaptation to new hosts and ecological settings.

Modularity in receptor-binding proteins (tail fibre, tail spike) as well as in endolysins has been extensively studied before, though to our knowledge it has not been systematically quantified and compared to other functional classes. Both receptor-binding proteins and lysins can play an important role in host range determination[67], and previous studies have repeatedly demonstrated their rapid evolution in the face

of adaptation to new hosts, particularly in receptor-binding proteins[30,68,69]. It is therefore not surprising that these proteins would have evolved a LEGO-like, modular architecture that facilitates rapid structural alterations to aid viral adaptation[7]. There are nevertheless important differences between the two groups in terms of how the resulting domain mosaicism has been and continues to be shaped by evolution. While receptor-binding proteins and endolysins are both specific in that they contain enzymes that recognise and hydrolyse specific sugar moieties, the diversity of the sugar repertoire on which they act can be quite different. Receptor-binding proteins often use surface polysaccharides as the primary receptor, notably capsular polysaccharides and LPS, which due to their rapid evolution can often vary considerably, even between two bacterial isolates of the same lineage[70]. This means that phages are under selective pressure to rapidly adapt to new hosts that may bear completely different surface receptors than their close relatives. A good example is *Klebsiella pneumoniae*, which is known to often exchange polysaccharide synthesis loci with other bacterial lineages[71] while its phages are known for not only extensive modularity of receptor-binding proteins[28] but also the existence of phages with complex tails with a broad host spectrum[72]. In line with this, we found clear evidence of the emergence of recent mosaicism within tail fibres and tail spikes.

Endolysins, on the other hand, target the peptidoglycan of their bacterial hosts. While there is a considerable diversity of peptidoglycans in bacteria[73], its diversity does not vary as dramatically between different lineages of the same species as can be the case with surface receptors. Consequently, one would expect a weaker diversifying selection acting within phages that infect closely related bacteria and a stronger one for those phages that infect distantly related hosts. In line with this reasoning, Oechslin and colleagues recently found that the fitness costs of endolysin exchange between phages increased for viruses infecting different bacterial strains or species[30]. However, they also found evidence of recombination-driven exchange of endolysins between virulent phages infecting the same host and the prophages carried by this host, pointing to the likely importance of recombination in driving the evolution of endolysins. This is consistent with the previous report of the extensive mosaic architecture of domains in endolysins in Mycobacteriophages[57] and across all Uniprot data[29]. Our results corroborate these findings by further showing that the diversification of phage endolysins via domain shuffling is an evolutionarily ongoing phenomenon.

Another major group for which there was evidence of extensive domain mosaicism were core replication proteins, particularly DNA polymerases. This result may seem counter-intuitive as core replication proteins are known to contain highly conserved sites due to the very precise way in which they process and metabolise DNA/RNA. But the DNA replication machinery is known to be highly diverse across the tree of life[74], including in viruses[52], and this diversity is known to have been evolving since the existence of the last common universal ancestor (LUCA) with evidence for the importance of recombination and domain shuffling in this process[51]. It can be thus expected that much of the domain mosaicism that we detect in this study is ancient and predates the emergence of bacteria and phages. However, there are a few arguments to suggest that such mosaicism has been emerging, and continuously emerges, during co-evolution between bacteria and phages. First, the scale of diversity of (and mosaicism in) some core replication proteins, for example, DNA polymerases or endonucleases, suggests that maintaining such diversity must have been beneficial for phages. Second, previous studies have reported the modularity of DNA polymerases[75] as well as the plasticity and modularity of the DNA replication machinery as a whole[76] in T4-like phages. The authors argued that such flexibility gives these viruses an edge in adapting to their diverse bacterial hosts[76]. Finally, our data points to clear examples of recently emerged domain mosaicism in DNA polymerases, ribonucleotide reductases and replication initiation proteins.

This suggests that core replication proteins continue to evolve in the process of bacteria-phage co-evolution.

One possible and potentially important driver of the diversity of core replication proteins in phages could be bacterial defence systems. There is a growing body of literature describing bacterial defence systems that target phage replication machinery to prevent viral infection and their spread in bacterial populations. One example is the DarTG toxin that was recently shown to ADP-ribosylate phage DNA to prevent phage DNA polymerase from replicating viral DNA and escape mutations in DNA polymerase allowed the phage to process the modified DNA[77]. Another example is the Nhi, a bacterial nuclease-helicase that competes with the phage DNA polymerase for the 3′ end of DNA to prevent phage replication[78]. A recent study by Stokar-Avihail and colleagues systematically investigated molecular mechanisms of phage escape from 15 different phage-defence systems in bacteria and found that such escape was often linked to mutations in core replication proteins including DNA polymerase, DNA primase-helicase, ribonucleotide reductase or SSB proteins[24]. The authors speculate that, from the evolutionary point of view, it makes sense for the bacterial defences to target essential components of the viral core replication machinery as an escape mutation would likely induce greater fitness cost for the virus. We thus think that the observed diversity and mosaicism observed within and between the proteins associated with core nucleotide metabolism reflects the ongoing co-evolutionary arms race between bacterial phage-defence systems and phages co-adapting to new bacterial defences. Given that mutations can often bear a high fitness cost, the recombination of existing folds could be a viable evolutionary mechanism of adaptation to move across the steep fitness landscape.

Altogether, our results can be viewed as one approach to identify evolutionary hotspots in phage genomes. Bacteria employ a wide range of, often highly genetically diverse, strategies to resist infection by phages and mobile genetic elements. Variation in how bacteria protect themselves over time, space and phylogeny means that no single strategy—or even a combination or strategies—can universally work for either side[79]. This is the type of scenario where one expects balancing selection (e.g., negative frequency-dependent selection) to maintain diversity of such strategies[80], and where genetic innovation can be evolutionarily favoured[81]. In this context, biological modularity enhances adaptability, allowing organisms to reconfigure or expand their functional repertoires in response to these changing environmental pressures[82]. Therefore, we would expect protein modularity, and the resulting domain shuffling, to become increasingly associated over time with functions that are essential in overcoming different bacterial resistance mechanisms that determine host range, such as host entry, lysis, and evasion of multiple bacterial defence systems.

An important aspect of ongoing protein evolution are applications in phage biotechnology and engineering, particularly in the realm of protein design. Understanding the fundamental building blocks of phage proteins and their functional implications could facilitate the rational design of synthetic phages with tailored functionalities[83,84] or modified host range[32,35]. For instance, insights into receptor-binding proteins, endolysins, and DNA polymerases could be leveraged to engineer phages that are more effective in targeting specific bacterial strains, thereby enhancing their utility in phage therapy[85] or diagnostics[86]. Additionally, the knowledge of how domains can be shuffled and diversified, powered by new AI-aided technologies like CADENZ[87], could inform the design of modular phage-based biosensors[88], delivery vectors[89] or enzybiotics[90]. This could be particularly useful in applications requiring high specificity and adaptability, such as targeted drug delivery or environmental monitoring. Furthermore, the identification of instances of recent diversification via domain shuffling suggests avenues for directed evolution experiments aimed at generating novel functionalities. Overall, our study not only advances our understanding of phage evolution but also provides a foundational framework that could be exploited for the rational design of phage-derived biotechnological tools.

As mentioned before, each of our two approaches to study domain mosaicism has strengths and weaknesses. While ECOD-based mosaicism is a robust approach to detect domain mosaicism, we showed that it is very restrictive and bound to miss genuine cases of protein fragment shuffling (e.g., horizontal swaps of homologous domains, sub-domain recombination or domain gain/loss), especially in proteins that are under-represented in domain databases. On the other hand, while sequence-based mosaicism is likely to identify these problematic cases, it can result in false-positive cases of mosaicism, for example stemming from highly variable rates of evolution in different areas of proteins. One good example are tail length tape measure proteins. They often exhibited mosaic signal by sequence but we were not able to confirm any genuine cases of the recent emergence of mosaicism either due to the presence of long and repetitive coiled coil regions or due to the occurrence of frequent splits of very long, near-identical ORFs into multiple ones. Ideally, a comprehensive understanding of the role of protein modularity in phage evolution would integrate sequence, domain, and structural information.

This leads us to an important caveat: our conclusions are most robust for well-represented and diverse protein functions, such as structural proteins or core replication proteins. This is a common challenge in studies of horizontal gene transfer and genetic recombination. While recombination promotes the emergence of diversity, greater diversity, in turn, makes it easier to detect composite sequences. Given the limitations discussed above, it is to be expected that a great reservoir of mosaicism exists beyond what was reported in this study, namely in (1) proteins of unknown functions, (2) proteins which are under-represented in domain databases and (3) in less frequent, accessory proteins that themselves could have emerged as a result of domain shuffling and diversifying selection acting on the phage pangenome. We thus expect that our results are only the tip of the iceberg which is the true extent of domain mosaicism in phage populations.

Our study also raises intriguing questions that extend beyond its current scope but offer fertile ground for future research. For example, our dataset is heavily skewed towards DNA phages, leaving the modularity of faster-mutating RNA phages largely unexplored. Another complex yet captivating question is whether phages are intrinsically more modular than bacteria. Addressing this would involve navigating challenges such as horizontal gene transfer which can blur the lines between bacterial and viral genes, database selection—ideally of temporal nature from co-evolving populations or bacteria and viruses—and the need for functionally comparable datasets. These questions, while methodologically demanding, could represent exciting directions for future research.

## Methods

### Data
We downloaded all complete bacteriophage genomes from NCBI Virus in January 2022 using the following criteria: virus = bacteriophage, genome completeness = complete, sequence type = RefSeq, yielding 4,548 complete genomes. We then detected open reading frames in those genomes using the approach based on `MultiPhate2`[91]. This resulted in 462,721 predicted protein sequences which were clustered with `mmseqs2`[92] (release 14-7e284) using the following parameters: minimum sequence identity = 0.3, sensitivity = 7, coverage = 0.95, yielding 133,624 clusters.

### HMM profile construction
For each of the clusters, a representative protein sequence was taken as the one suggested by `mmseqs2` (i.e., sequence with the most alignments above the special or default thresholds with other sequences of

the database and these matched sequences). Of those, 50 included more than 10 unknown characters and were thus excluded from further analysis. Each of the remaining 133,574 representative sequences was then used as a starting point to build a hidden Markov model (HMM) profile for each of the clusters. The profile was built by aligning the UniClust30 database[93] (release `UniRef30_2020_06_hhsuite`) against each representative sequence with `hhblits` with the following parameters: minimum probability = 90%, minimum sequence identity with master sequence = 10%, minimum coverage with master sequence = 30%, and other parameters set to default[40]. If no hit was found, the HMM consisted of one single (representative) sequence which was used as the initial point to further `hhblits` searches. The resulting profiles are referred to as rHMMs (HMM profiles of representative proteins) throughout this work. See also Supplementary Fig. S1 for a visual outline of the methodology.

## All-by-all profile-profile comparison

All 133,574 rHMMs were pairwise compared using `hhblits`, with a minimum hit probability threshold set at 0.5 and all other parameters left at their default settings. For each pair of rHMMs, we performed two types of alignments: query-to-subject and subject-to-query, as HMM-HMM alignments are non-commutative. For each alignment type, we calculated both query and subject coverage as the proportion of the aligned region length to the total length of the query and subject sequences, respectively. Hit probability was conservatively calculated as the minimum of the probabilities obtained from both types of alignments; percentage identity was similarly determined as the minimum of the percentage identities from both alignments. Unless specified otherwise, only hits that met or exceeded a (final) minimum hit probability of $p \geq 0.95$ were considered. To assign rHMMs into protein families, we recalculated hit probability, query and subject coverage as above but restricting the homology search to $p \geq 0.95$ instead of 0.5 Then we only considered all pairs of rHMMs with a pairwise coverage cov = min(qcov,scov) $\geq$ 0.8. For each pair, we then calculated a weighted score of $p \times$ cov, which was used as a weight of a undirected network. Finally, we used a Markov clustering algorithm (MCL)[94] with an inflation factor `-I 2` to cluster rHMMs into 72,078 families.

## Functional annotation

To assign each rHMM to a functional category, we used the Prokaryotic Virus Remote Homologous Groups database (PHROGs; version 4)[41]. Every rHMM was compared with the PHROGs HMM profile database using `hhblits`. We used functional classes as defined by PHROGs, but we additionally simplified and merged the names referring to closely related biological functions (e.g., *RusA-like Holliday junction resolvase* and *RuwC-like Holliday junction resolvase* became *Holliday junction resolvase*; *Dda-like helicase* and *DnaB-like replicative helicase* became *DNA helicase*; *head-tail adaptor Ad1* and *head-tail adaptor Ad2* became *adaptor*, etc.). The exact mapping of used functional categories onto PHROGs is provided in Supplementary Data S1. Only functional classes that (1) were assigned to PHROGs with the total number of at least 500 sequences and (2) were found in at least 20 rHMMs were considered (unless stated otherwise). Additionally, every rHMM was compared with a database of antidefence proteins[42] using hhblits, and those that had hits to PHROGs and some specific antidefence function were assigned the specific antidefense class. Functional classes were assigned as those with hits to a known class at 80% coverage and 95% probability hit threshold. Finally, rHMMs with hits to more than a single functional class were discarded unless they only co-occurred with generic classes like tail or structural protein.

## Domain detection

To detect domains in rHMMs, we used the Evolutionary Classification of Protein Domains[43] database (ECOD, version from 13.01.2022). Each

of the 133,574 rHMMs was compared to the `HHpred` version of the ECOD database using `hhblits` with a minimum probability of 20% and otherwise default parameters. Domains were considered as those hits to rHMMs with probability $p \geq 0.95$ and subject coverage scov $\geq 0.7$.

## Detection of mosaic protein pairs

To look for potential mosaicism between rHMMs at the domain level (cf., 2A), we searched for pairs of rHMMs that shared a domain of the same topology (i.e., fold; ECOD T-groups), detected at the 95% probability threshold, while each containing domains that belonged to different ECOD X-groups (i.e., there is absence of evidence of homology between these domains at both sequence and structural level). To look for potential mosaicism between proteins at the sequence level (cf., Fig. 4B), we compared all rHMMs with each other at the permissive probability threshold of 50% to account for potential distant homology between the two sequences. Again, the query and subject coverage were calculated as the total number of residues in the aligned sequence regions by their respective lengths. The pair of rHMMs was considered mosaic if it was found to share a similar genetic fragment (probability $p \geq 0.95$ percentage identity $p_{id} \geq 0.3$) in the background of the absence of homology at the permissive probability threshold: max(scov,qcov) $\leq 0.5$. We only considered rHMM pairs with a minimum aligned fragment length of 50aa.

## Statistics and reproducibility

Multiple ECOD domains were tested for being over-represented in rHMMs with evidence of domain mosaicism. Additionally, all considered functional classes were tested for being over-represented in families of rHMMs with evidence of domain mosaicism. All these tests were done using one-tailed Fisher's exact tests and Bonferroni correction for multiple testing, as described below.

**Odds ratio to be over-represented in proteins with evidence of domain mosaicism.** Each domain architecture was classified as mosaic (i.e., having evidence of mosaicism) or non-mosaic (no evidence of mosaicism). Then for each topology (ECOD T-group) we calculated the number mosaic domain architectures including this topology ($m_t$), number of mosaic domain architectures not including this topology ($m_{nt}$), number of non-mosaic domain architectures including this topology ($n_t$) and number of non-mosaic domain architectures not including this topology ($n_{nt}$). Then the odds ratio was calculated as:

$$\text{OR} = \frac{m_t/m_{nt}}{n_t/n_{nt}}. \tag{1}$$

**Functional classes over-represented in families with evidence of domain mosaicism.** Each rHMM family was classified as mosaic by our ECOD-based or sequence-based definition of domain mosaicism if it contained any rHMM with a respective mosaic signal. Then, for each of the 99 well-specified functional classes (excluding the generic classes 'tail' and 'structural protein') we counted the number of mosaic families (by ECOD and sequence, respectively) that include at least one rHMM assigned to this functional class $n_m$ and the number of mosaic families (by ECOD and sequence, respectively) that do not include any rHMM assigned to this family but include at least one rHMM assigned to any of the remaining 98 functional classes, $m_m$. Likewise, for each functional class, we counted the number of non-mosaic families that include at least one rHMM assigned to this functional class, $n_n$, and the number of non-mosaic families that do not include any rHMM assigned to this functional class but include at least one rHMM assigned to any of the remaining 98 functional classes, $m_n$. Those numbers were then passed to a contingency table to perform the one-tailed Fisher's exact test. As the test was performed separately for each of the 99 functional classes, we adjusted the $p$-values with a Bonferroni correction.

## Taxonomic and ecological metadata

We first assigned a bacterial host (Genus level) to a phage genome using information from the NCBI database (release October 2022). We used the 38th release of the ICTV classification[95] to assign both Genus and Family. (It is worth noting that many phage genomes lacked assigned taxonomic classifications, particularly at the family level.) For lifestyle prediction, we employed BACPHLIP[96] (version 0.9.6). Phages with a temperate probability of ≥90% were classified as temperate and those with a temperate probability of ≤10% were classified as virulent. Overall, host, family, genus and lifestyle information was assigned to $n = 4543$, $n = 1579$, $n = 3611$ and $n = 3375$ genomes, respectively.

Finally, we assigned host, taxonomic family, genus, and lifestyle to rHMMs involved in mosaic pairs. Specifically, if all proteins within a given rHMM originated from the same group (excluding unclassified or unknown ones), the rHMM was allocated to that group. However, numerous rHMMs consisted of proteins that were conserved across multiple groups, including different families, genera, hosts, or lifestyles. In such instances, we were unable to assign them to any specific group.

## Reporting summary

Further information on research design is available in the Nature Portfolio Reporting Summary linked to this article.

## Data availability

Genomes used in this study were downloaded from NCBI Refseq in January 2022. Representative HMM profiles were built with UniClust30 release `UniRef30_2020_06_hhsuite`. Functional annotation was carried out with PHROGs version 4. Domain architectures were predicted using ECOD version `ECOD_F70_20200207`. All data necessary to reproduce the results have been uploaded to `FigShare` and are accessible at https://doi.org/10.6084/m9.figshare.24004092, with NCBI accession numbers which can be used to link all protein sequences to the corresponding genomes. Domain architecture lookup in different functional classes is available at: https://bognasmug.shinyapps.io/PhageDomainArchitectureLookup. Source data are provided with this paper.

## Code availability

Two computational pipelines were used to generate the results: phage-protein-modularity-data and phage-protein-modularity-figures. The first pipeline was written in Python3 and processes NCBI RefSeq data and carries out HMM profile construction, all-by-all HMM comparison and HMM-HMM comparison to PHROGs and ECOD databases. The second pipeline, written in R version 4.1, takes the output of the first pipeline as input and generates all outputs and figures presented in the publication. Both pipelines are available under the following Github releases: https://github.com/bioinf-mcb/phage-protein-modularity-data, archived under https://doi.org/10.5281/zenodo.10021838, https://github.com/bioinf-mcb/phage-protein-modularity-figures, archived under https://doi.org/10.5281/zenodo.10026778.

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

## Acknowledgements

This work was financed by the Polish National Agency for Academic Exchange (B.S., R.M.), the Polish National Science Centre OPUS Grant (grant 2020/37/B/NZ8/03492; B.S., K.S., R.M.), the EMBO Installation Grant (B.S., K.S., R.M.) and First TEAM programme of the Foundation for Polish Science, co-financed by the European Union under the European Regional Development Fund (grant POIR.04.04.00-00-5CF1/18-00; S.D.-H.). We thank Virkam Alva, Carina Büttner, Alan R. Davidson, Mart Krupovic, Andrei Lupas, Joana Pereira and Eugen Pfeifer for helpful discussions and/or comments related to this work which was instrumental in finalising this manuscript. The open-access publication of this article was funded by the Priority Research Area BioS under the program 'Initiative of Excellence–Research University' at the Jagiellonian University in Krakow.

## Author contributions

Original idea: R.M. Conceptualisation: B.S., R.M. Data Collection: B.S., K.S. Methodology: B.S., K.S., S.D.-H., R.M. Software Development: B.S., K.S., R.M. Data Analysis: B.S., E.P.C.R., S.D.-H., R.M. Visualisation: B.S., R.M. Writing—Original Draft: B.S., R.M. Writing—Review & Editing: B.S., K.S., E.P.C.R., S.D.-H., R.M. Supervision: R.M. Funding Acquisition: R.M. Project Administration: R.M.

## Competing interests

The authors declare no competing interests.
