## [Peer Review File · Nature Communications]

Ongoing shuffling of protein fragments diversifies core viral functions linked to interactions with bacterial hostsREVIEWER COMMENTS

Reviewer #1 (Remarks to the Author):

This paper investigates modularity and horizontal gene transfer in a large collection of phage-related proteins. To this end, they used HMM-HMM proteins and compared them with functionally annotated domains of the ECOD database. They convey general modularity in the different molecular functions of phages and highlight certain interesting cases, such as DNA polymerases, endolysins, and receptor-binding proteins.

I enjoyed reading this paper, and it is topical in phage (and general protein) research. The study is well executed and convincing, holistic while exploring several specific protein classes and cases. I found the methodology clear and most figures illuminating. The end discussion seems fine, though I might miss some perspectives on phage biotechnology and engineering.

I wonder if HMM-HMM profiles are the best way of detecting exchanged motifs in proteins? Could the authors elaborate on this choice with some (recent) literature and suggest potential alternatives?

What is unclear is how the HMM-HMM are compared. Can the authors give some explanation for this or clearly point to a reference elucidating this? (it is explained in the methods, though it might be briefly discussed in the main text)

One missing aspect is a study of how these findings differ for the taxonomic and ecologic subgroups of the phages. I would like to see this discussed and potentially illustrated with some (supplementary) results.

Will you make your data available (e.g., the domains) via an open repo/database so others can build upon this (e.g., dryad)? It seems to be a great resource for protein engineering.

The following papers might be of interest:

<https://royalsocietypublishing.org/doi/10.1098/rsif.2018.0595>
<https://www.biorxiv.org/content/10.1101/2022.09.17.508230v2>
<https://www.pnas.org/content/102/39/13773>
<https://www.sciencedirect.com/science/article/pii/S0079610717301347>

How does this relate to the PhaLP database (Criel et al., 2021)? These discuss the modularity of phage lytic proteins at length.

- p2 "Very good 20 examples are receptor-binding proteins (including tail fibres, tail spikes) or endolysins": sentence might be improved
- figure 1 can be strongly improved for publication, i.e., removing numbers in the x-axis, adjusting the background. The right bar might be coloured according to molecular category? I suggest a manual cleanup using graphical software (e.g. Inkscape)
- I think Fig 2A is not entirely clear and can be improved for clarity (reading the text helps). replacing T- and H- group with topology and homology would help.
- Figure S2 (green and red) is not color-blind friendly, better use (e.g.) blue and orange
- p13, l3 & l4: "which" => "that"
- p13, l23: "Here, we" (comma)
- p16, l14: "fibres, which"

Reviewer #2 (Remarks to the Author):

Smug et al. investigate protein modularity in phages using HMM profiles and sequence- or domain-based homology search. They find evidence for frequent past and ongoing protein mosaicism, particularly in DNA polymerases, tail fibres, tail spikes and endolysins, and suggest different mechanisms for protein diversification. Further, they argue that this mosaicism is the result of the co-evolutionary arms race between phages and their bacterial hosts.

The paper is well-written and the analysis is exhaustive. Protein modularity in phages has not previously been explored at such a large scale and the manuscript provides interesting new insights. As such, I only have a few minor comments.

Minor comments:

Title: the first part is fine but 'to tackle bacterial resistance mechanisms' seems to be more speculation (although the arguments are very convincing) than one of the main results, so I would remove it from the title

p.2, l.26: it could be better defined what 'true extent of protein modularity' means here

p.13, l.25: there is a 'second' but no 'first'

p.14, l.4: something seems to be wrong or missing in the phrase 'likely co-occur many out of all possible combinations'

p.19, l.18: are the cases representative of the respective functional classes or of the mechanisms?

p.20, l.14: no 'a'

p.22, l.22 no 'the'

p.24, l.3: the transition here seems to be a bit abrupt?

Fig.3: the rHMM description seems to be missing for the tail fibre protein in the last line?

Fig.S10: there are inverted ? in the legend?

Did you check if patterns of modularity differ for proteins that come from lytic or temperate phages?

Reviewer #3 (Remarks to the Author):

This manuscript presents an analysis on within-protein modularity in phage genes. Although examples of this have been known, this large-scale analysis shows that it is a wide-spread phenomenon and can be linked to anti-defense mechanisms. Overall, the methodology is sound, the results are well represented and interpreted and the paper is very well written and mostly easy to follow. Nevertheless, I have some comments.

1. In the introduction it sounds like "protein mosaicism" and "protein modularity" refer to the same phenomenon. I find it a little confusing to have two different words for the same most important concept of the paper. The authors could make this equivalence explicit, or only use one only, or (if applicable) state the difference.

2. p.6 l.8 It is not clear what "topology" means in this context. It would also be interesting to know how evolutionary relatedness and structural homology have been determined. Of course, readers can look up the original paper. But since the concepts are highly relevant for the manuscript, they should be explained to a level that the manuscript can be followed.

3. As I understand figure 1, it shows which functional classes have which domains, but not all the domains in one line need to be present in the same protein. Nevertheless, in p. 7 it is stated "For example, functional classes such as exonucleases, endonucleases, DNA polymerases or endolysins each contained as many as 4 distinct H-groups, each found in at least 5 rHMMs (see Figure 1)." It is currently not obvious how this information could be seen in Figure 1.

4. p.10 last paragraph. What is the difference between "contemporary mosaicism" and "recently emerged mosaicism"? Before, only contemporary mosaicism is described and it is unclear what recently emerged mosaicism in this concluding paragraph refers to.

5. p. 17 "This relationship was qualitatively identical when we subtracted all rHMM pairs where we detected domain mosaicism with ECOD" This sentence is unclear, which relationship is meant here and what does "qualitatively identical" mean?

6. The paper generally lacks statistics. E.g., functional classes where modularity was found are described. Are these classes overrepresented in modular rHMMs compared to the remaining? Such analyses could make the paper even stronger.

7. The paper focusses on functions of these modular proteins which is certainly a very interesting aspect. But other questions immediately come to mind. Are some kinds of phages more prone to having modular proteins depending on taxonomy, life style (virulent/temperate), RNA or DNA genomes? Although such analyses might go beyond the scope of the paper, the authors could comment on this in the discussion.

8. Another very interesting further question is - are phages really more prone to modularity? How does this level of modularity described here compare to bacterial proteins? Is there anything known from the literature?

Additional points:

Define abbreviations when first mentioning them (e.g., HMM)

Figure 5. What do the colors in the right part of the plot mean?

p. 25 The representatives were searched against UniClust30. It should be stated if each representative had a hit or what happened if no hit was found.

Figure S10. Typo in legend ("?=")

Table S1 has some formatting (bold, color) and it is unclear what this means.

REVIEWER COMMENTS

Reviewer #1 (Remarks to the Author):

This paper investigates modularity and horizontal gene transfer in a large collection of phage-related proteins. To this end, they used HMM-HMM proteins and compared them with functionally annotated domains of the ECOD database. They convey general modularity in the different molecular functions of phages and highlight certain interesting cases, such as DNA polymerases, endolysins, and receptor-binding proteins.

I enjoyed reading this paper, and it is topical in phage (and general protein) research. The study is well executed and convincing, holistic while exploring several specific protein classes and cases. I found the methodology clear and most figures illuminating.

We would like to thank the reviewer for the kind words and for time spent carefully reading our word and the resulting, very helpful feedback.

The end discussion seems fine, though I might miss some perspectives on phage biotechnology and engineering.

Response: We appreciate your suggestion to include perspectives on phage biotechnology and engineering. In response, we have expanded the Discussion section to incorporate insights into how our findings could be applied in these fields, particularly in protein design and phage engineering (page 29). We believe this addition enriches the manuscript by providing a broader context for our results.

I wonder if HMM-HMM profiles are the best way of detecting exchanged motifs in proteins? Could the authors elaborate on this choice with some (recent) literature and suggest potential alternatives?

Response: We have elaborated on our choice of using HMM-HMM profiles for detecting exchanged motifs in the Discussion section (page 26, lines 6-13). While HMM-HMM comparisons are currently the state-of-the-art method for detecting homology with high sensitivity, we acknowledge that there are emerging techniques, such as those based on natural language processing. However, these newer methods have not yet been as widely tested as HMM-HMM comparisons, which have proven to be highly reliable.

What is unclear is how the HMM-HMM are compared. Can the authors give some explanation for this or clearly point to a reference elucidating this? (it is explained in the methods, though it might be briefly discussed in the main text)

Response: We appreciate the suggestion for clarity. We have now included a brief explanation of the HMM-HMM comparison process in the beginning of the Results section with a corresponding citation (page 4, lines 8-13), while retaining the detailed methodology in the Methods section. This should make the approach more accessible to readers who may not delve into the Methods section.

One missing aspect is a study of how these findings differ for the taxonomic and ecologic subgroups of the phages. I would like to see this discussed and potentially illustrated with some (supplementary) results.

Response: Thank you for highlighting the importance of examining the taxonomic and ecological dimensions of our study. In response to your suggestion, which echoes other suggestions made by the two remaining reviewers, we have conducted extensive additional analyses to explore how our findings differ across various taxonomic and ecological subgroups of phages.

Specifically, we found that temperate phages exhibit a higher proportion of mosaic genes compared to lytic phages, aligning with existing literature that suggests more frequent horizontal gene transfer in temperate phages. We also observed significant variations in the proportion of mosaic genes between phages assigned to different taxa (family, genus), infecting different bacterial hosts (genus level) and different lifestyles (temperate/virulent). Interestingly, we found that domain mosaicism, particularly recently emerged mosaicism, often transcends taxonomic and ecological borders. These new insights have been incorporated into the manuscript in the form of a new section in the Results section (pages 24-25) as well as seven new supplementary figures S14-S20. We believe these additions enrich the depth of our study and provide a more nuanced understanding of protein modularity in phages.

Will you make your data available (e.g., the domains) via an open repo/database so others can build upon this (e.g., dryad)? It seems to be a great resource for protein engineering.

Response: Yes, we fully intend to make our data publicly available upon publication. Specifically, we plan to fully comply with the policy of the journal and upload all data necessary to reproduce the results to Figshare:

<https://figshare.com/s/d7838adde00826fb6fa0>

We have also made our data available directly from the webserver:

bognasmug.shinyapps.io/PhageDomainArchitectureLookup.

We appreciate the suggestion that this dataset could be a valuable resource for the community, particularly for those interested in protein engineering, and we have taken it seriously.

The following papers might be of interest:

<https://royalsocietypublishing.org/doi/10.1098/rsif.2018.0595>

<https://www.biorxiv.org/content/10.1101/2022.09.17.508230v2>

<https://www.pnas.org/content/102/39/13773>

<https://www.sciencedirect.com/science/article/pii/S0079610717301347>

Response: Thank you for suggesting these papers. We have reviewed and cited them (references 4,7,8,85) in the revised manuscript to strengthen our discussion and contextualise our findings. Specifically, we have now restructured the Introduction to include the concept of biological modularity and relate it to protein modularity.

How does this relate to the PhaLP database (Criel et al., 2021)? These discuss the modularity of phage lytic proteins at length.

Response: Thank you for mentioning the PhaLP database by Criel et al., 2021. While PhaLP focuses on the domain architectures of endolysins, our study extends this by using ECOD domains to identify proteins with shared and non-homologous domains. This allowed us to confirm that the observed mosaicism in endolysins likely results from horizontal evolution. Our findings thus complement and enrich the insights provided by the PhaLP database. We now refer to the PhaLP database in the manuscript in parts of the text which refer to endolysins in the Results (page 15, lines 2-4) and in the Discussion (page 27, lines 24-27).

- p2 "Very good 20 examples are receptor-binding proteins (including tail fibres, tail spikes) or endolysins": sentence might be improved

Response: We have now improved this sentence for clarity.

- figure 1 can be strongly improved for publication, i.e., removing numbers in the x-axis, adjusting the background. The right bar might be coloured according to molecular category? I suggest a manual cleanup using graphical software (e.g. Inkscape)

Response: We have now improved this Figure according to the reviewer's suggestions.

- I think Fig 2A is not entirely clear and can be improved for clarity (reading the text helps). replacing T- and H- group with topology and homology would help.

Response: We have now improved this Figure according to the reviewer's suggestions.

- Figure S2 (green and red) is not color-blind friendly, better use (e.g.) blue and orange

Response: We have now improved this Figure according to the reviewer's suggestions.

- p13, l3 & l4: "which" => "that"

Response: We have now corrected this.

- p13, l23: "Here, we" (comma)

Response: We have now corrected this.

- p16, l14: "fibres, which"

Response: We have now corrected this.

Reviewer #2 (Remarks to the Author):

Smug et al. investigate protein modularity in phages using HMM profiles and sequence- or domain-based homology search. They find evidence for frequent past and ongoing protein mosaicism, particularly in DNA polymerases, tail fibres, tail spikes and endolysins, and suggest different mechanisms for protein diversification. Further, they argue that this mosaicism is the result of the co-evolutionary arms race between phages and their bacterial hosts.

The paper is well-written and the analysis is exhaustive. Protein modularity in phages has not previously been explored at such a large scale and the manuscript provides interesting new insights. As such, I only have a few minor comments.

We appreciate the reviewer's positive remarks on our manuscript and are pleased that the reviewer found our analysis exhaustive and insightful. We have carefully considered the comments and have made the appropriate revisions to address them.

Minor comments:

Title: the first part is fine but 'to tackle bacterial resistance mechanisms' seems to be more speculation (although the arguments are very convincing) than one of the main results, so I would remove it from the title

Response: We agree with the reviewer that the title is speculative. We have now changed it to "*Ongoing shuffling of protein fragments diversifies core viral functions linked to interactions with bacterial hosts*" which we believe represents the main findings more accurately.

p.2, l.26: it could be better defined what 'true extent of protein modularity' means here

Response: We have now rephrased the sentence where we changed "true extent of protein modularity" into "the extent to which domain mosaicism occurs in phages and its relationship to genetic and functional diversity in phages" (page 3, lines 11-13).

p.13, l25: there is a 'second' but no 'first'

Response: We have now corrected this.

p.14, l4: something seems to be wrong or missing in the phrase 'likely co-occur many out of all possible combinations'

Response: We have now corrected this.

p.19, l18: are the cases representative of the respective functional classes or of the mechanisms?

Response: They constitute representative functions in which we found such examples. We have now rewritten this paragraph to avoid potential confusion (page 21, lines 22-30).

p.20, l14: no 'a'

Response: We have now corrected this.

p.22, l22 no 'the'

Response: We have now corrected this.

p.24, l3: the transition here seems to be a bit abrupt?

Response: We have smoothed out the transition to make the flow of the text more natural.

Fig.3: the rHMM description seems to be missing for the tail fibre protein in the last line?

Response: This was an alignment problem which we have now corrected.

Fig.S10: there are inverted ? in the legend?

Response: We have now corrected this.

Did you check if patterns of modularity differ for proteins that come from lytic or temperate phages?

Response: Yes we did. In fact, given that this comment was independently raised by all three reviewers, we have now added an entire section on the relationship of domain mosaicism with phage taxonomy and ecology (pages 24-25). Relating to this specific question, our results now show that temperate phages have a higher proportion of mosaic genes compared to lytic ones and that recently emerged mosaicism can sometimes be linked to exchanges between temperate and lytic phages.

Reviewer #3 (Remarks to the Author):

This manuscript presents an analysis on within-protein modularity in phage genes. Although examples of this have been known, this large-scale analysis shows that it is a wide-spread phenomenon and can be linked to anti-defense mechanisms. Overall, the methodology is sound, the results are well represented and interpreted and the paper is very well written and mostly easy to follow.

We thank the reviewer for the constructive feedback on our manuscript. We appreciate the positive comments and have addressed all concerns in the revised version.

1. In the introduction it sounds like "protein mosaicism" and "protein modularity" refer to the same phenomenon. I find it a little confusing to have two different words for the same most important concept of the paper. The authors could make this equivalence explicit, or only use one only, or (if applicable) state the difference.

Response: We appreciate this comment as it highlights a potential source of confusion for other researchers in the field. We have now clarified the confusion by (i) explicitly defining 'protein modularity' in the context of biological modularity and phage biology, (ii) defining 'domain mosaicism' and using it instead of 'protein mosaicism' and (iii) avoiding the term 'protein mosaicism' altogether (see pages 2-3). In short, in the newly submitted manuscript, protein modularity refers to the organisation of a protein into functional domains or units while domain mosaicism refers to the outcome of such a phenomenon.

2. p.6 l.8 It is not clear what "topology" means in this context. It would also be interesting to know how evolutionary relatedness and structural homology have been determined. Of course, readers can look up the original paper. But since the concepts are highly relevant for the manuscript, they should be explained to a level that the manuscript can be followed.

Response: We agree with the reviewer that this is another potential source of confusion. In the new version of the manuscript we have now clearly explained what 'topology' is (unique arrangement of secondary structures in the protein domain) and are now avoiding the dubious term 'structural homology'. We have now rewritten this part of the Results (page 7, lines 8-17).

3. As I understand figure 1, it shows which functional classes have which domains, but not all the domains in one line need to be present in the same protein. Nevertheless, in p. 7 it is stated "For example, functional classes such as exonucleases, endonucleases, DNA polymerases or endolysins each contained as many as 4 distinct H-groups, each found in at least 5 rHMMs (see Figure 1)." It is currently not obvious how this information could be seen in Figure 1.

Response: Indeed, the domains shown in Figure 1 are not necessarily found within the same rHMM. We have now modified the sentence to make it clear Furthermore, to make it even more transparent, we have added a column to Figure S10 which shows the maximum number of H-groups detected per rHMM.

4. p.10 last paragraph. What is the difference between "contemporary mosaicism" and "recently emerged mosaicism"? Before, only contemporary mosaicism is described and it is unclear what recently emerged mosaicism in this concluding paragraph refers to.

Response: The main difference between them is that 'contemporary mosaicism' is defined via ECOD-based mosaicism with an additional restriction of the minimum percentage identity of 50% for the shared fragment between two proteins. In contrast, 'recently emerged mosaicism' refers to proteins sharing fragments of minimum percentage identity of 70% (high-confidence) and 90% (very-high confidence). We have now added more explicit definitions in the text (page 2, lines 2-14).

5. p. 17 "This relationship was qualitatively identical when we subtracted all rHMM pairs where we detected domain mosaicism with ECOD" This sentence is unclear, which relationship is meant here and what does "qualitatively identical" mean?

Response: To make the sentence clearer, we rephrased it and referred to the specific Figure S12B where two scenarios can be visually compared (page 18, lines 1-4).

6. The paper generally lacks statistics. E.g., functional classes where modularity was found are described. Are these classes overrepresented in modular rHMMs compared to the remaining? Such analyses could make the paper even stronger.

Response: Thank you for your insightful comment on the need for more statistical rigour in our paper. We have taken your suggestion to heart and made several updates to enhance the statistical robustness of our findings. Specifically, we have revised Figure 4 to highlight functional classes that are significantly overrepresented among mosaic rHMMs. These are now marked with an asterisk and were identified using Fisher's exact test, with p-values adjusted for multiple testing using the Bonferroni correction.

Additionally, we would like to draw your attention to other figures where statistical analyses have been applied:

- In Figure 2c and 2d, only domains significantly overrepresented in mosaic rHMMs are displayed, based on Fisher's exact tests with Bonferroni correction.
- Figure S5 now includes a correlation test to assess the relationship between domain frequency and diversity.
- Figure S11 has been updated to include both the correlation and the results of a correlation test, focusing on the relationship between structural and genetic diversity.
- We have also included two Supplementary Tables S2 and S3 where we list the exact odds ratios and p-values for each ECOD domain tested for being overrepresented in mosaic rHMMs (Figures 2c-2d) and each functional class tested for being overrepresented in mosaic rHMMs (Figure 4).

7. The paper focusses on functions of these modular proteins which is certainly a very interesting aspect. But other questions immediately come to mind. Are some kinds of phages more prone to having modular proteins depending on taxonomy, life style (virulent/temperate), RNA or DNA genomes? Although such analyses might go beyond the scope of the paper, the authors could comment on this in the discussion.

Response: We agree that these are intriguing questions. As they echo similar questions raised by other reviewers, we have now substantially expanded the Results by adding a new section (pages 24-25) and new Supplementary Figures S14-S20 to explore the relationship of domain

mosaicism with phage taxonomy and ecology (please refer to our reply letter to the Editor, reply to Reviewer #1's question).

As for the question about DNA vs RNA phages, we also agree that it is very interesting, however our dataset contains only 35 phages classified as RNA, making them too under-represented to conduct any meaningful analysis. However, we now mention this question in a new paragraph in the Discussion regarding potential new directions of research in the field.

8. Another very interesting further question is - are phages really more prone to modularity? How does this level of modularity described here compare to bacterial proteins? Is there anything known from the literature?

Response: Thank you for raising this intriguing question about the comparative modularity between phage and bacterial proteins. We fully agree that this is a compelling avenue for future research, but it also presents a series of complex challenges that make it difficult to address within the scope of our current study.

First, the pervasive nature of horizontal gene transfer between bacterial and phage genomes complicates the clear delineation of gene origins. While databases like CheckV could offer some resolution, a significant number of genes would still remain ambiguously classified. Second, selecting an appropriate database that fairly represents both bacterial and phage proteins is non-trivial. A focus on prophages within bacterial genomes, for example, would inherently exclude lytic phages and likely underrepresent ancient mosaicism. Likewise, temporal, high-quality genomic data from co-evolving populations of bacteria and phages are difficult to obtain.

Most critically, a fair comparison would necessitate the examination of functionally analogous proteins between bacteria and phages. While bacteria have a range of proteins with varying degrees of modularity, only a limited set of functions are shared with phages, making a comprehensive comparison challenging. One could for example compare the modularity of DNA related families. But that would narrow down the analysis and introduce a bias in favour of phages because we already know their DNA related proteins are very modular.

Given these complexities, we believe that a thorough investigation of this question would constitute a long-term project in its own right. To the best of our knowledge, the literature has yet to delve into this specific comparison, further underscoring the need for dedicated study. To highlight the complexity and importance of this question, we are now briefly discussing it in the context of the new paragraph in the Discussion about potential new directions of research in the field (page 30, last paragraph).

Additional points:

Define abbreviations when first mentioning them (e.g., HMM)

Response: Abbreviations like HMM have been defined upon first mention.

Figure 5. What do the colors in the right part of the plot mean?

Response: We agree that this is confusing. We have now unified the colours to green (proteins with a functional hit) and grey (proteins with unknown function). It has also been explained in the Figure legend.

p. 25 The representatives were searched against UniClust30. It should be stated if each representative had a hit or what happened if no hit was found.

Response: If no hit was found, the rHMM consisted of one single (representative) sequence that was used as the initial point to further hhblits searches. We have now added a clarification in the Methods (page 32, lines 18-19).

Figure S10. Typo in legend ("?=")

Response: We have now corrected this.

Table S1 has some formatting (bold, color) and it is unclear what this means.

Response: We have now removed it as it was superfluous.

REVIEWERS' COMMENTS

Reviewer #1 (Remarks to the Author):

I thank the authors for their thorough revision. I particularly liked the additional parts regarding ecological niches and biotechnological potential. The manuscript seems to be ready to be accepted.

Congratulations

Reviewer #2 (Remarks to the Author):

The authors have addressed the reviewer comments appropriately, which improved the clarity of the manuscript and added substantial new analyses about the ecological and taxonomic dimensions of protein modularity. This manuscript presents a comprehensive and well-presented study that provides interesting new findings on protein modularity and phage evolution.

Reviewer #3 (Remarks to the Author):

I have thoroughly reviewed the author's answers and the revised version of the manuscript. I enjoyed reading it again and I am happy to report that the authors did a good job in addressing the comments and suggestions I made in my previous review. The changes made to the manuscript have significantly improved its clarity and overall quality. The incorporation of an additional section on ecology has also enriched the paper.